

# Earth observation informed modelling of flash floods

C. Scott Watson[1], Maggie Creed[2], Januka Gyawali[3], Sameer Shadeed[4], Jamal Dabbeek[4], Divya L. Subedi[5], Rojina Haiju[5]

[1] COMET, School of Earth and Environment, University of Leeds, LS2 9JT, UK
5  [2] James Watt School of Engineering, The University of Glasgow, Glasgow, G12 8QQ, UK
[3] Practical Action, Kathmandu, Nepal
[4] Civil and Architectural Engineering Department, An-Najah National University, Nablus, Palestine
[5] Water and Climate Program, Nepal Development Research Institute (NDRI), Lalitpur, Nepal

*Correspondence to*: C. Scott Watson (c.s.watson@leeds.ac.uk)

10  **Abstract**

More frequent extreme rainfall events in a changing climate increase the risk of flash flooding that is affecting populations globally. However, the flood hazard modelling required to reduce disaster risk in populated urban environments is often limited by the availability of data required for model calibration and validation. In this study, we use a historical flood event captured by 5 m resolution satellite imagery to quantify the effects of flood model complexity and inform flood hazards under future 15  climate scenarios in the West Bank, Palestine. Flooding in January 2013 affected over 12,500 people and large areas of cropland. Vegetation loss and damage during the event were captured using satellite imagery and a normalised difference vegetation index (NDVI), and used as a reference flood extent. The physics-based HEC-RAS flood model best reproduced this NDVI-derived inundation extent (F1 score = 0.76), although the FastFlood model was able to produce a similar inundation pattern (F1 score = 0.74) over 300 times faster. Simulated flood depths from both models were similar; FastFlood displayed a 20  mean difference of -0.03 m and a mean absolute error of 0.51 m when compared to HEC-RAS. Climate analysis revealed that the January 2013 rainfall corresponded to a historical return period of between 1 in 5 and 1 in 10 years. In comparison, a 1 in 100-year rainfall event (RX1day (maximum 1-day precipitation) of 148 mm) based on historical data (1985–2014) could increase by almost 40% (to 205 mm) in the mid-future (2041–2060), which could cause 23% (4 km$^2$) greater inundation compared to the 2013 event. Although the patterns of future precipitation in the region are uncertain, our flood hazard maps 25  can support urban planning and infrastructure development to manage storm water runoff, particularly where ephemeral channels, or wadis, intersect the road network.

## 1 Introduction

A warming climate with more frequent extreme rainfall events (Min et al., 2011; O'Gorman, 2015) coupled with increased exposure of populations and infrastructure to flooding (Alfieri et al., 2017; Jongman et al., 2012; Tellman et al., 2021) is driven 30  by factors including higher magnitude flood events (Hirabayashi et al., 2013; Slater et al., 2021a), human-modified catchment runoff characteristics (Kundzewicz et al., 2014), and encroachment into flood-prone areas (Andreadis et al., 2022; Devitt et



al., 2023). The Intergovernmental Panel on Climate Change's Sixth Assessment Report (IPCC AR6) states that in many parts of Asia, the risks related to climate change are projected to increase progressively for 1.5°C, 2°C, and 3°C of global warming (Shaw et al., 2023). Vulnerability to flooding, which is a function of physical, social, and economic factors, is generally highest

in developing countries and informal settlements that lack planning and infrastructure to manage flood water (De Risi et al., 2013; Kron, 2005). Therefore, pro-poor risk-informed planning is essential to reduce flood risk in an equitable way for future developments (Galasso et al., 2021). However, robust flood modelling processes required to inform disaster risk reduction strategies require high-quality input data including future precipitation trends generated by analysing historical precipitation observations alongside global climate models (Cannon et al., 2015; Shrestha et al., 2023); an accurate digital elevation model

that represents the channel and floodplain topography (Hawker et al., 2018; Muthusamy et al., 2021; Watson et al., 2024); and data to calibrate and validate model outputs (Di Baldassarre et al., 2009; Molinari et al., 2019). In developing countries, one or more of these inputs are often lacking This can force simplifications to the modelling process and choice of flood model complexity, which can subsequently limit the effectiveness of model outputs in decision making. For example, global flood hazard maps at ≥90 m resolution that use open access DEMs can provide valuable probabilistic hazard information at regional

scales (Dabbeek et al., 2020; Sampson et al., 2015). However, model intercomparisons highlight inconsistencies that are often linked to digital elevation model (DEM) resolution and accuracy, which affects how local channel and floodplain complexities are represented (Hawker et al., 2018; Jenkins et al., 2022; McClean et al., 2020; Trigg et al., 2016).

Earth observation data have broad applicability in flood disaster response and planning, owing to the large spatial coverage,
frequent revisit times, and diversity of information provided by different sensors. For example, information on the duration and intensity of storms is available from precipitation monitoring missions such as NASA's Global Precipitation Measurement (Huffman et al., 2023; Pradhan et al., 2022); the height of rivers and inundated land can be derived using satellite altimetry (Asadzadeh Jarihani et al., 2013; Zakharova et al., 2020); antecedent conditions including soil moisture can be obtained using radar (Entekhabi et al., 2010; Kornelsen and Coulibaly, 2013); flood routing and the effects on vegetation can be quantified
using changing reflectance characteristics (Cian et al., 2018; Shrestha et al., 2013); and inundation extents can be mapped using both optical and radar data (DeVries et al., 2020; Di Baldassarre et al., 2009; Mateo-Garcia et al., 2021; Schumann et al., 2018). Barriers to using earth observation datasets for city-scale flood hazard modelling include their availability and accessibility since it is only within the last decade that spatial and temporal resolution has improved, alongside open access licensing. Therefore, the use of earth observation data in historical flood assessments is still limited in data-sparse areas,
particularly to capture infrequent flash flooding.

Reducing flood disaster risk requires knowledge of current and future flood hazards under realistic climate scenarios, combined with the ability to influence decision making at local scales. The Tomorrow's Cities project was designed to respond to this challenge through a risk-informed urban planning approach (Decision Support Environment (DSE)), weighted towards
benefitting marginalised and vulnerable communities (Galasso et al., 2021). In this study, we aimed to draw on our experience



in the application of the DSE in Nablus, Palestine, to evaluate established and emerging flood modelling approaches to demonstrate their applicability in a data-sparse flash flood environment. Our objectives were to: (1) quantify the impact of an extensive historical flood event using pre- and post-flood satellite imagery; (2) evaluate three flood hazard models of increasing complexity using the historical event for validation; and (3) use this to inform an assessment of current and future flood hazard in the region.

## 2 Study Area

Our study focused on the north-western part of the West Bank, Palestine (in Jenin and Tulkarm governorates), which is thought to exhibit the greatest flood hazard, due to high rainfall and runoff potential (Shadeed, 2018)(Figure 1). In this region, increases in extreme rainfall, impermeable surfaces, and lack of infrastructure to deal with flood water have led to increasing vulnerability to flash floods (Asmar et al., 2021; Hassan et al., 2010; Hawajri et al., 2016; Shadeed, 2018). A particularly damaging flood event followed heavy and sustained winter rainfall in early January 2013, which affected 12,500 people and caused widespread damage to agricultural land (Hawajri et al., 2016; OCHA, 2013). The flooding highlighted existing vulnerabilities and the importance of advancing disaster risk reduction strategies and community and government levels (OCHA, 2013).





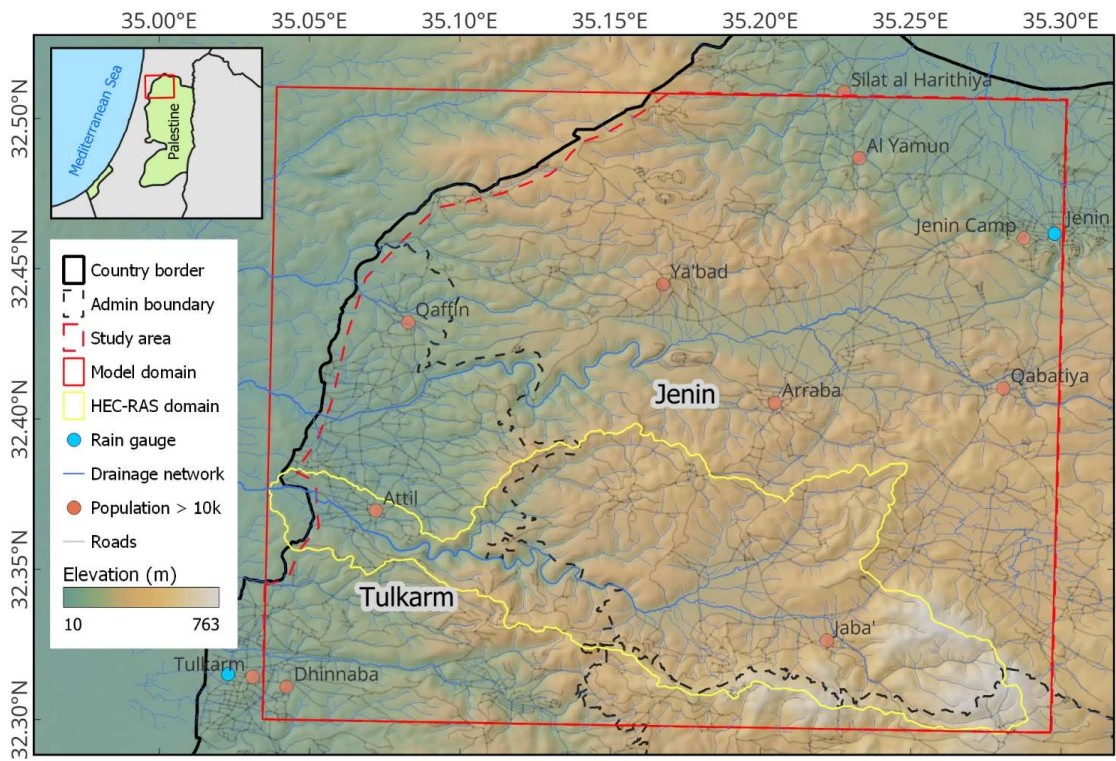


**Figure 1: Study area spanning the Jenin and Tulkarm administrative governorates in the West Bank, Palestine. Rain gauge stations in Jenin and Tulkarm are shown. The Nablus rain gauge station is 9 km to the south of the study area.**





## 3 Methodology

### 3.1 Digital elevation model (DEM) preparation and analysis


A 10 m resolution DEM was created to underpin the flood hazard modelling. Contour lines at 1 m vertical intervals were supplied by the Ministry of Local Government (Palestine) covering the study area and the *Topo to Raster* tool in ArcGIS Pro 3.0 was used to interpolate a digital elevation model at 10 m resolution. The method and date of acquisition of the contour lines were not known. Therefore, we compared a hillshade of the DEM to satellite imagery in Google Earth to estimate an acquisition date of 2015–2016, based on the construction date of large buildings that were apparent in the DEM. Differencing the 10 m custom DEM from the 30 m Copernicus Digital Elevation Model (GLO30) revealed spatially variable offsets and artefacts (Figure 2a). Therefore, five components of the DEM were defined for independent adjustments to improve the DEM before flood hazard modelling ([1] to [5] shown in Figure 2a). Component [1] extended the custom DEM beyond the study area to avoid boundary effects in the flood modelling and was filled with the GLO30 DEM. Components [2, 3 and 5] were coregistered to the GLO30 DEM independently to correct their systematic offsets using a blockwise coregistration pipeline in the xDEM Python package, which incorporated a bias correction, iterative closest point registration, followed by the coregistration of Nuth and Kääb (2011). Component [4] was replaced with the GLO30 DEM due to the presence of systematic artefacts. The refined custom DEM had a normalised median absolute deviation (NMAD) of 0.67 m when differenced with the GLO30 DEM, compared to 2.22 m before adjustment (Figure 2). Components [1] and [4] were excluded from this NMAD calculation since these were areas filled with the GLO30 DEM. Hydrological conditioning was applied to the coregistered 10 m custom DEM using the *BreachDepressionsLeastCost* tool in Whitebox 1.4.0 (Lindsay, 2016) with a maximum breach distance of 1 km. A stream network was then derived using a flow accumulation threshold of 1,000 cells, which was selected based on a visual inspection of stream sources using the Google Satellite Imagery basemap in QGIS.






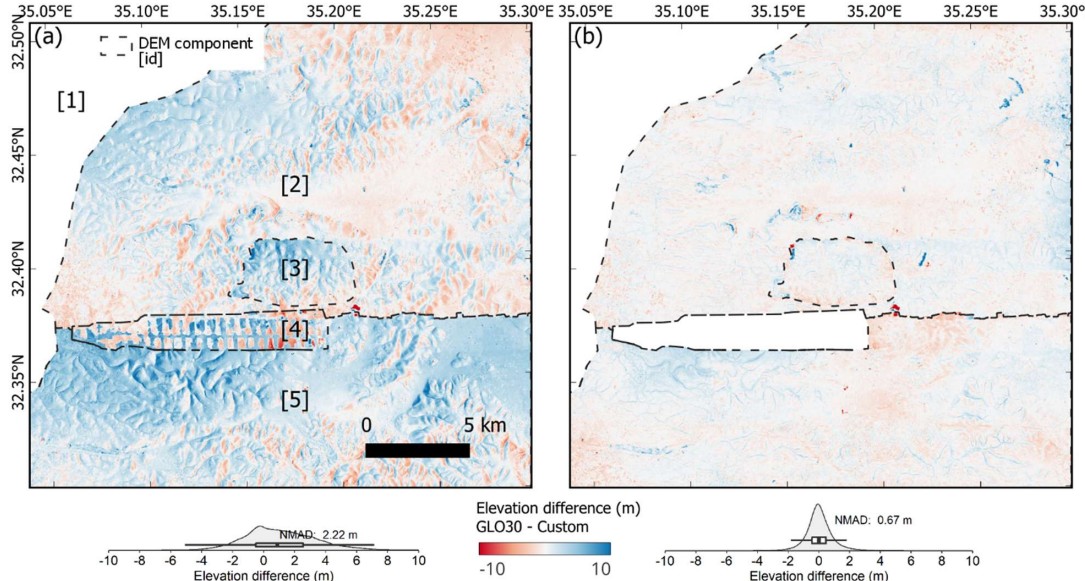

**Figure 2: (a) DEM of difference between the GLO30 DEM and the custom DEM derived in this study from contour data. Labelled components are discussed in the text. (b) DEM of difference post-coregistration to the GLO30 DEM.**

### 3.2 Earth observation data

### 3.2.1 RapidEye imagery

Pre- and post-flood availability of RapidEye satellite imagery was used to assess the flood extent of the flooding in January 2013. Two multi-spectral 5 m resolution RapidEye-1 satellite images (31 December 2012 and 15 January 2013) processed to surface reflectance (L3A) were accessed from PlanetLabs. The 2013 image was co-registered to the 2012 image using Cosi-Corr (S. Leprince et al., 2007) and ENVI 5.6.3 to remove a spatially variable misregistration (east-west mean offset: 6.4±6.3 m, north-south mean offset: 4.2±6 m) using a second order polynomial. A normalised difference vegetation index (NDVI) (1) was derived for the two acquisitions using the near infrared and red bands.

$$NDVI = \frac{(Nir - Red)}{(Nir + Red)} \tag{1}$$

High chlorophyll reflectance in the Nir band compared to low reflectance in the red band means the NDVI is an indicator of vegetation presence and health (Pettorelli et al., 2005; Tarpley et al., 1984). The difference in NDVI values pre- and post-flood can be used to reconstruct the inundated extent using the change in reflectance of the damaged or scoured vegetation, which would be observed as an NDVI decrease (Atefi and Miura, 2022; Miles et al., 2018). The short timespan between the RapidEye



acquisitions of our study side (15 days) meant that changes in NDVI values were expected to primarily correspond to the effects of the extreme rainfall and flooding on vegetation. To mask out insignificant change in NDVI, we manually digitised

'stable' sample polygons over areas where the spectral reflectance was less likely to be affected by the flooding, such as woodland and bare ground, and extracted their NDVI difference values pre- and post-flood. We then masked pixels with NDVI difference values less than two times the standard deviation of these stable polygons (NDVI<0.06) and sieved the output to retain connected clusters of at least 18 pixels (450 m$^2$). Finally, we intersected the remaining NDVI changes with the stream network derived in Section 3.1 to determine whether they were likely the consequence of fluvial flooding. The Shreve stream

order (Shreve, 1966) was allocated to the stream network derived in (3.1) and these values were used to sample the NDVI difference to investigate the relationship between stream order and the pre- and post-flood magnitude of NDVI change.

### 3.2.2 Land cover

ESA WorldCover 10 m v200 (Zanaga et al., 2021) was used to quantify the landcover of each area of significant NDVI change (3.2.1). We quantified this for both the full study area (506.5 km$^2$)(Figure 1), which included NDVI changes corresponding to

pluvial and fluvial flooding and agricultural activity, and separately for the NDVI changes directly connected to the stream network (Figure 1), which were most likely to be caused by fluvial flooding. To evaluate potential inundation impacts, we also used building footprints from the Global ML Buildings dataset (Microsoft, 2024), and the transportation network including roads and tracks from OCHA (OCHA, 2021), which were more complete than OpenStreetMap data.

### 3.3 Rainfall data and climate scenarios

Rainfall data for the January 2013 flood event were available from the Palestine Meteorological Department for Jenin (JEN00001, 145 m elevation), Tulkarm (TUL00002, 7 m elevation), and Nablus (NAB00003, 73 m elevation) stations (Figure 1, 3a). Inverse distance weighted interpolation was used to create a grid representing peak rainfall (mm/hour) from the station data on 8[th] January (Figure 3b). We also downloaded the total rainfall on the 8[th] January 2013 from the calibrated GPM L3 IMERG V06 precipitation product (Huffman et al., 2019) for comparison with the gauge stations (Figure 3b).





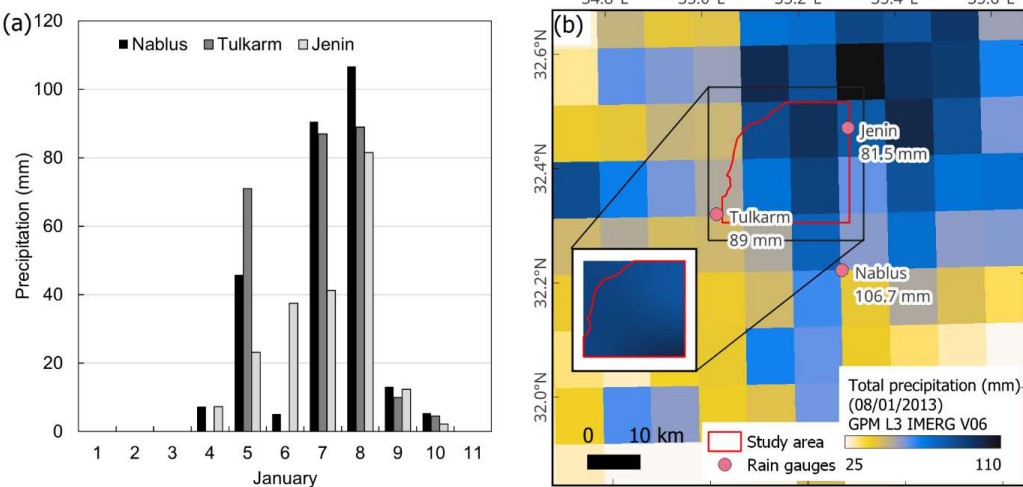

**Figure 3: Rainfall data for the January 2013 flood event. (a) Rainfall recorded at three rain gauges bounding the study area. (b) Rainfall derived from the GPM L3 IMERG V06 product on the 8th of January 2013 and interpolated rainfall between the three rain gauges for the study area (inset). Labelled values correspond to the total recorded rainfall on 8th January 2013.**

Historical and future projected rainfall data were used in a climate change analysis. Historical rainfall data recorded at Nablus
station (NAB00003) (9 km south of the study area) from 1985–2014 were provided by the Palestine Meteorological
Department. Coupled Model Intercomparison Project Phase 6 (CMIP6) Global Circulation Models (GCMs) are used for the
future projected rainfall data. The GCMs selected for this study are GFDL-ESM4 and MPI-ESM1-2-HR, as recommended by
Hamed et al. (2022) and Mesgari et al. (2022). Hamed et al. (2022) compared CMIP5 and CMIP6 models over the Middle East
and North African (MENA) region using historical simulations and future projections, while Mesgari et al. (2022) provides an
assessment of CMIP6 models' performances and projection of rainfall based on Shared Socio-economic Pathways (SSPs)
scenarios over Middle East, North Africa, Afghanistan, and Pakistan (MENAP). The two SSPs considered in this study are
SSP2-4.5 (medium challenges to mitigation and adaptation), and SSP5-8.5 (high challenges to mitigation, low challenges to
adaptation). The time period of 1985–2014 is considered as historical. The time periods of 2021–2040, 2041–2060 and 2081–
2100 are classified as near future, mid-future, and far future respectively, in accordance with IPCC AR6.

We analysed the historical rainfall patterns and characteristics, with focus on RX1day (annual maximum 1-day rainfall) (e.g.
Shrestha et al., 2023). First, the GCM data were bias corrected and statistically downscaled to the desired spatial and temporal
resolution. The bias correction and statistical downscaling were undertaken using the empirical quantile mapping method,
which maps the probability distribution of rainfall of GCMs with the probability distribution of the observed rainfall. The
concept of quantile mapping can be understood as:

$$X_{future,t}^{corr} = inverse\ ecdf\ _{reference}^{obs}\left(ecdf_{reference}^{Model}\left(X_{future,t}^{Model}\right)\right) \qquad (1)$$



Here, *ecdf* is the empirical cumulative distribution function for the reference time period, $X_{future,t}^{Model}$ is the raw GCM at time *t* in the future, $ecdf_{reference}^{Model}$ is the empirical cumulative distribution function of the GCM for the reference period, $inverse\ ecdf\ _{reference}^{obs}$ is the inverse empirical cumulative distribution function of the observed rainfall for the reference

period, and $X_{future,t}^{corr}$ is the corrected estimate of $X_{future,t}^{RModel}$. The monthly *ecdf* for this study was developed using the observed rainfall data at Nablus station and GCM hindcast data for the period 1985-2014. Gudmundsson et al., (2012) illustrates this procedure in detail. Stationary or non-stationary rainfall frequency analysis was then performed on the bias corrected RX1day values based on the Mann Kendall trend test, to quantify the rainfall value associated with a given return period (Milly et al., 2008; Slater et al., 2021b). The temporal disaggregation of rainfall values from the rainfall frequency analysis was done using

the Global Precipitation Measurement (GPM) - Integrated Multi-satellitE Retrievals for GPM (IMERG) (GPM-IMERG) of half hourly temporal resolution and 0.1° X 0.1° spatial resolution based on the highest flood event recorded in Nablus.

### 3.4 Flood modelling

### 3.4.1 January 2013 flooding

Three models were used to simulate the January 2013 flooding and evaluate their performance speed and accuracy: FastFlood

v0.12 is a new computationally efficient model that has shown good agreement with fully dynamic physics-based models whilst requiring up to 1,500 times less computation time (van den Bout et al., 2023); HAIL-CAESAR is a high performance version of the Caesar-Lisflood model that uses simplified shallow-water equations (Coulthard et al., 2013); and HEC-RAS 6.4.1 is a physics-based hydraulic model capable of 2D unsteady flow simulations using Shallow Water Equations. The data input and parameters for the flood models are shown in Table 1. Manning's roughness values were uniformly applied across

the study area for each simulation. Model sensitivity to the roughness value was determined by increasing the value from 0.01 to 0.07 in 0.01 increments and assessing the modelled flood extent for depths greater than 0.1 m against the NDVI-derived reference data (Section 3.4.2). The FastFlood and Hail-Caesar models were run for the model domain shown in Figure 1 and the outputs were then clipped to the study area to avoid artefacts at the model boundary. A smaller study catchment was used for the HEC-RAS simulations to make them computationally viable, which was subsequently used as the reference area to

compare the outputs from all three flood models.

**Table 1. Flood model input parameters.**

| Model | DEM | Rainfall data | Settings | Manning's Roughness values tested |
|---|---|---|---|---|



| FastFlood | 10 m resolution (Section 3.1) clipped to the study area (Figure 1) | 8th January 2013 interpolated grid (mm/hr) (Section 3.3, Figure 4b) | Solver: very high quality | 0.01–0.05 |
|---|---|---|---|---|
| HAIL-CAESAR | | 8th January 2013 hydro index of three rainfall zones (mm/hr) for 24 hours | See the example parameter file in the supplement | 0.01–0.05 |
| HEC-RAS | 10 m resolution (Section 3.1) clipped to HEC-RAS domain (Figure 1) | 8th January 2013 interpolated grid (Section 3.3, Figure 4b) (mm/hr) for 24 hours | Downstream boundary: normal depth Computation interval: 10 seconds Equations: SWE-ELM | 0.01–0.07 |

### 3.4.2 Model comparison

Modelled flood depth was evaluated against the NDVI changes that intersected with the stream network, which indicated the
removal or damage of vegetation during the 2013 flood (Section 3.2.1). FastFlood and HAIL-CAESAR were run across the
full study area (Figure 1), whereas HEC-RAS was run for a smaller catchment due to computational limitations. Therefore,
the models were evaluated across these two domains. Since the NDVI changes were not expected to be fully representative of
the observed flood extent, for example on banks lacking vegetation, polygons of NDVI decrease that intersected with the
stream network and that visually appeared to correspond to the 2013 flood event (Section 3.2.1) were manually selected. These
polygons were manually edited in some cases to improve their representation of the inundation extent (e.g. Figure S1b). Buffers
of 110 m were then created from both sides of the stream centreline and these were clipped to encompass the NDVI polygons
to form validation areas (Figure S1). These areas were then used to derive accuracy assessment scores for each flood model.
The F1 score (2), which is a weighted average of precision (ratio of the true positive modelled flood area to the total modelled
flood area) and recall (ratio of true positive modelled flood area to the total reference (NDVI) flood area), was used to represent
overall model accuracy on a 0–1 scale where 1 is the highest accuracy (Kabir et al., 2020; Konapala et al., 2021). Additionally,
the intersection over union (IoU) ratio (3) was used to quantify the amount of overlap between the predicted flood extent and
the NDVI reference extent.





$$F1\ score = \frac{(\text{Precision} \times \text{Recall})}{(\text{Precision} + \text{Recall})/2} \tag{2}$$


$$IoU = \frac{\left(\begin{array}{c}\text{Intersecting area of the predicted flood extent}\\ \text{and the reference flood extent}\end{array}\right)}{\left(\begin{array}{c}\text{Combined area of prediced flood extent}\\ \text{and the reference flood extent}\end{array}\right)} \tag{3}$$

## 4 Results

### 4.1 January 2013 flooding

Peak rainfall during the 2013 flooding occurred on the 8[th] of January with 106.7, 89.0, and 81.5 mm of rainfall recorded at Nablus, Tulkarm, and Jenin rain gauges respectively (Figure 3). The GPM L3 IMERG V06 satellite product showed a maximum of 96.1 mm total rainfall in the study area for the same day, which was observed closest to Jenin station (Figure 3b).

### 4.1.1 NDVI change

The effects of the January 2013 flood event were evident in decreased NDVI values across the study area, particularly
corresponding to the stream network and a large area of ponded water that accumulated during the storm (Figure 4). NDVI changes were not exclusively confined to the stream network and incorporated the effects of pluvial flooding and seasonal agricultural activity. The RapidEye images spanned a fifteen-day window, and the post-flood image was captured five days after the peak rainfall, therefore the effects of vegetation change unrelated to the storm event were minimised. However, NDVI increases northeast of Arraba town were related to specific agricultural activities (Figure 4b). It was noted from Sentinel-2
data that NDVI values are typically increasing across the study area between the months of December–January and that this is also a time of crop harvest (Figure S2). There was a clear non-linear relationship between NDVI decrease and increasing Shreve stream order (Figure 4d). Except for stream orders 0–20, the median NDVI decrease for other streams exceeded the expected uncertainty of 0.06, with stream orders 200–220 displaying the largest median NDVI decrease of -0.33 (Section 3.2.1). The highest stream orders displayed greater spread in the NDVI change (Figure 4d), likely due to a combination of their
less ephemeral nature, greater carrying capacity, and lower detection of NDVI changes for channels in built-up environments.

The total area of NDVI decrease and increase across the study area was 65.3 km$^2$ (12.9% of the study area) and 12.4 km$^2$ (2.5% of the study area) respectively (Figure 5a), with the cropland land cover class displaying the greatest extent of NDVI decrease and NDVI increase with 13.2 km$^2$ (24% of all cropland) and 6.1 km$^2$ (11% of all cropland) of land affected respectively. The
NDVI decrease intersecting with the stream network, which was most likely to be a direct result of flooding, totalled 35.1 km$^2$ (6.9% of the study area) and cropland was the class most affected (10.5 km$^2$, 19% of all cropland) followed by grassland (9.4



km², 4.1% of all grassland) (Figure 5b). The area of NDVI decrease also included 1.8 km² (3.5%) of the built-up area (Figure 5b). Non-fluvial NDVI decrease was greatest for grassland (12.6 km², 5.5% of all grassland) (Figure 5c)

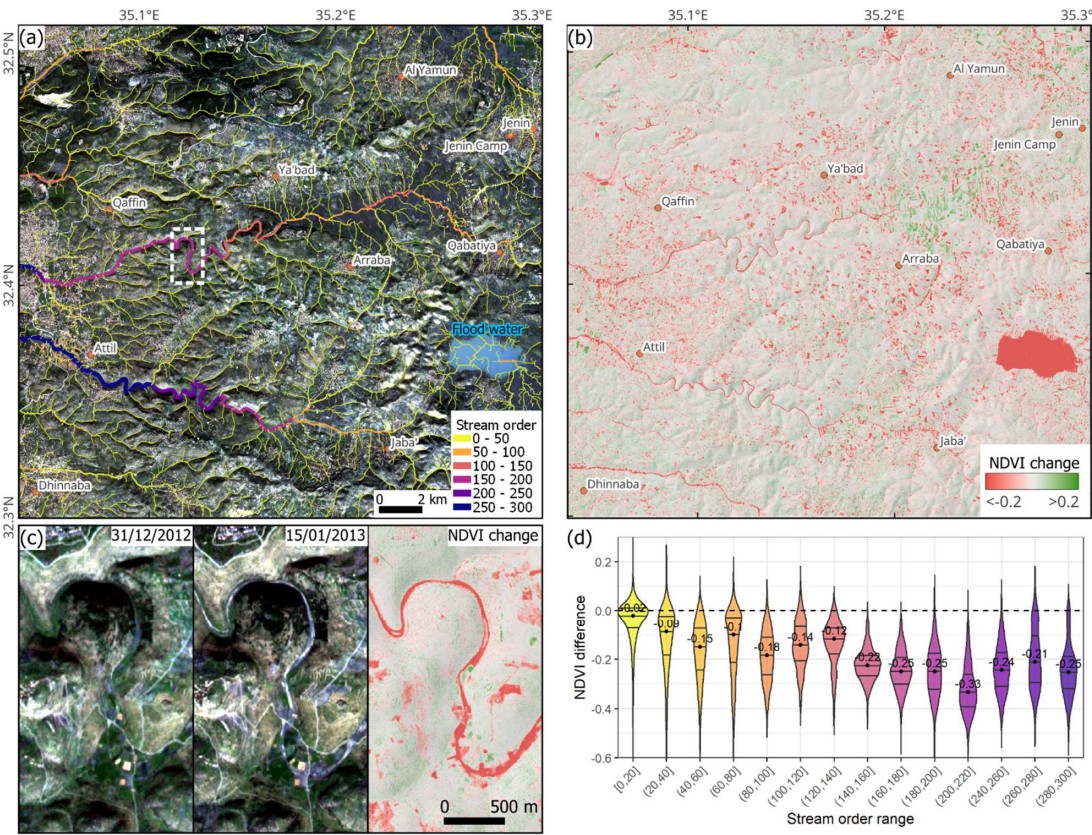

**Figure 4: (a) Shreve stream order in the study area. (b) NDVI change derived from pre- and post-flood RapidEye imagery. (c) Example of the pre- and post-flood RapidEye imagery with the corresponding NDVI change. (d) NDVI difference plotted with increasing stream order. Basemap (a and c) is a RapidEye-1 image (15th January 2013). Image © 2013 Planet Labs PBC.**



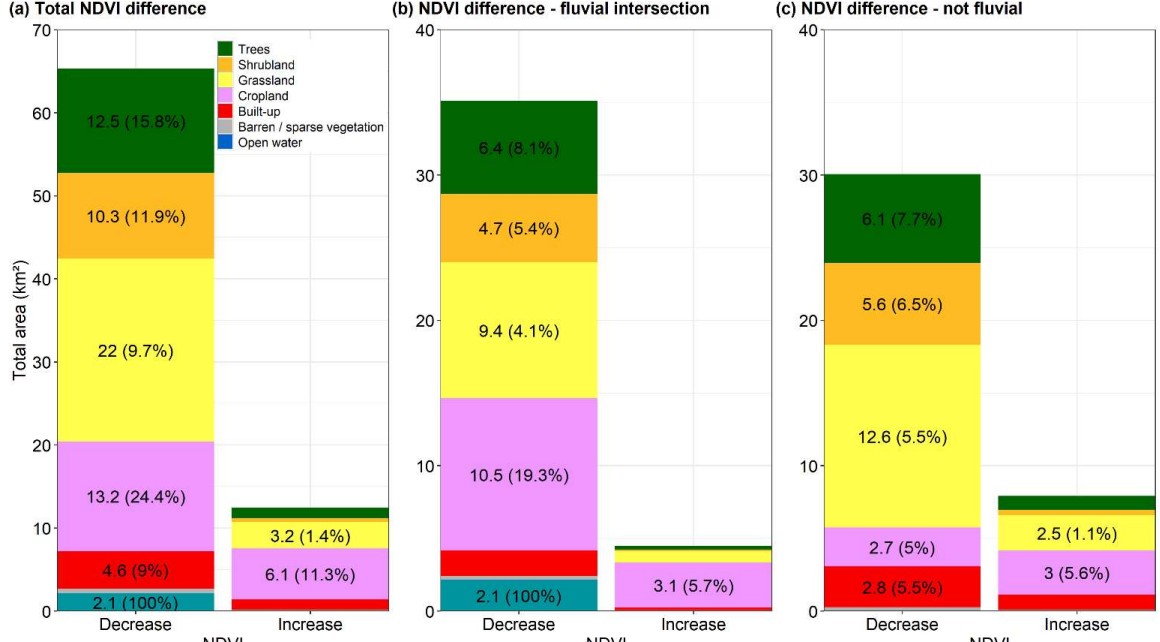

**Figure 5: (a) NDVI increase and decrease in the study area coloured by the corresponding land cover extracted from ESA World Cover v200. (b) NDVI increase and decreases for areas connected to the stream network and (c) for areas not connected to the stream network. The area of each class in square kilometres is labelled. Percentages represent the class area as a percentage of the total area of each land cover class in the study area. Bars are coloured by their land cover class.**

### 4.1.2 Flood model comparison

The modelled inundation extent across the full study area was 21.84 km$^2$ for FastFlood and 17.42 km$^2$ for HAIL-CAESAR for the 2013 flood event (Table 2). For the smaller HEC-RAS domain, the inundated areas were 3.05 km$^2$, 2.99 km$^2$, and 4.80 km$^2$ for FastFlood, HAIL-CAESAR, and HEC-RAS respectively (Table 2). FastFlood and HAIL-CAESAR were compared across the full study area, and all three models were directly compared in the HEC-RAS study domain (Table 2). Here, the F1 accuracy scores were 0.74, 0.75, and 0.76 for FastFlood, HAIL-CAESAR, and HEC-RAS respectively (Table 2). Similarly, the HEC-RAS model displayed the highest IoU score (0.61), although FastFlood (0.59) and HAIL-CAESAR (0.60) were similar (Table 2). The Manning's *n* values used in these models were 0.03, 0.02, and 0.06 respectively and the sensitivity of model accuracy to Manning's *n* values was low in the tested range of 0.01–0.06 (Table S1). The modelled flood depths from HAIL-CAESAR were a closer match to HEC-RAS compared to FastFlood (Figure 6). Here, FastFlood depths had a mean difference of -0.03 m and a mean absolute error of 0.51 m when compared to HEC-RAS, compared to 0.21m and 0.36 m respectively for HAIL-CAESAR (Figure 7).





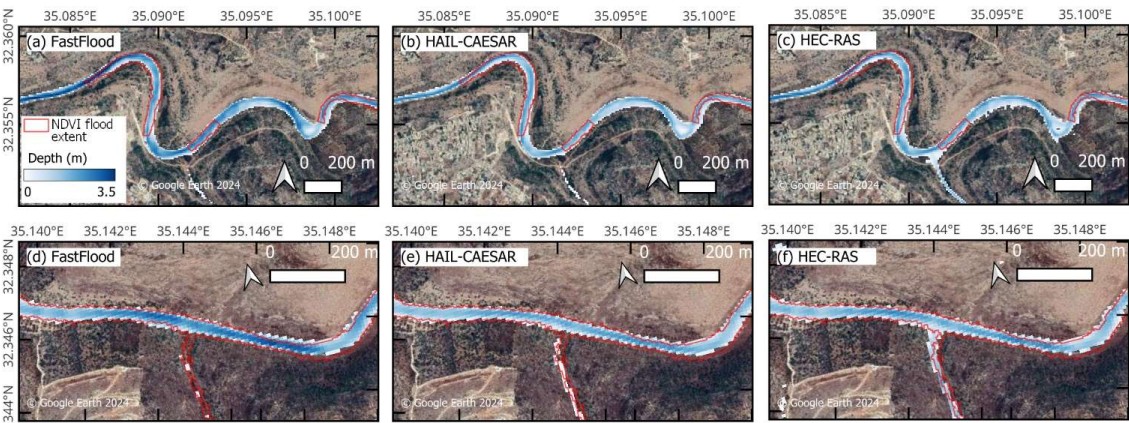

**Figure 6: Example modelled flood depths for the January 2013 rainfall event at two locations. NDVI-derived validation extents are shown as red polygons.**

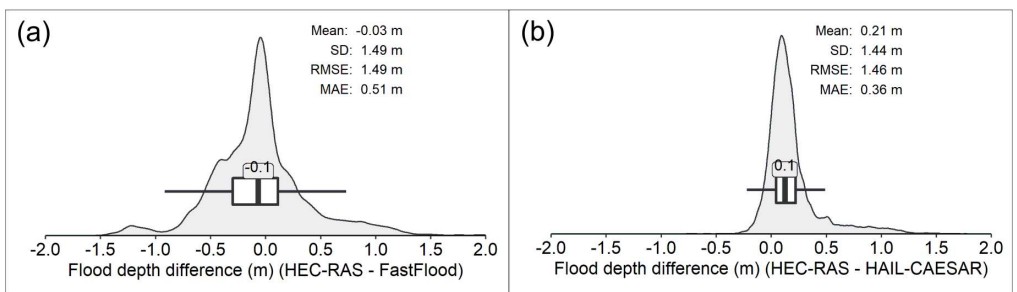

**Figure 7: Flood depth difference boxplots (median value is labelled) and half violin plots. (a) HEC-RAS flood depth minus FastFlood. (b) HEC-RAS flood depth minus HAIL-CAESAR. Mean, standard deviation (SD), root mean square error (RMSE), and mean absolute error (MAE) are shown.**

**Table 2. Flood model accuracy assessment for the 2013 flood compared to the NDVI-derived inundation extents.**

| Extent | Model | Mannings | F1 score | Precision | Recall | IoU | Inundated area (km$^2$) | Model runtime |
|---|---|---|---|---|---|---|---|---|
| Study area | FastFlood | 0.02 | 0.72 | 0.70 | 0.73 | 0.56 | 21.84 | ~40 seconds[1] |
| | HAIL-CAESAR | 0.03 | 0.73 | 0.75 | 0.71 | 0.57 | 17.42 | ~7 hours[2] |
| HEC-RAS domain | FastFlood | 0.03 | 0.74 | 0.73 | 0.76 | 0.59 | 3.05 | |
| | HAIL-CAESAR | 0.02 | 0.75 | 0.75 | 0.75 | 0.60 | 2.99 | Clipped from full study extent |
| | HEC-RAS | 0.06 | 0.76 | 0.68 | 0.86 | 0.61 | 4.80 | ~4 hours[1] |



---

[1] Desktop PC with 14 cores (3.3 Ghz). [2] High Performance Computing node with 40 cores (2.0 Ghz)

---

### 4.2 Future climate and flood hazard

### 4.2.1 Climate

Biases in the long term mean monthly rainfall between observed rainfall at Nablus rain gauge and the selected GCMs (Figure 8a) were observed. Although the GCMs were able to capture the monthly rainfall pattern, significant biases were observed between the observed and modelled datasets. Figure 8b shows the historical RX1day with future bias corrected RX1day for the two climate models and SSP scenarios we tested. A maximum increase of 18% and 24% with respect to the historical RX1day was seen in GFDL-ESM4 and MPI-ESM1-2-HR, respectively, for the SSP2-4.5 near future period (2021-2040) (Table

3). A negative change in RX1day, contrary to the majority of other locations worldwide (Arias et al., 2021), was observed in SSP5-8.5 scenario in the mid-future, whereas no changes in RX1day can be observed in SSP5-8.5 scenario in the near future period.

**Table 3: Percentage changes in future RX1day after bias correction**

| RX1day (mm) and % changes: GFDL-ESM4 | | | | | | | |
|---|---|---|---|---|---|---|---|
| Scenarios | Historical 1985–2014 | Near future 2021–2040 | % change | Mid-future 2041–2060 | % change | Far future 2081–2100 | % change |
| SSP2-4.5 | 71 | 84 | 18 | 72 | 1 | 82 | 15 |
| SSP5-8.5 | | 76 | 8 | 68 | -4 | 80 | 13 |
| RX1day (mm) and % changes: MPI-ESM1-2-HR | | | | | | | |
| SSP2-4.5 | 71 | 88 | 24 | 83 | 17 | 82 | 15 |
| SSP5-8.5 | | 71 | 0 | 75 | 6 | 73 | 3 |






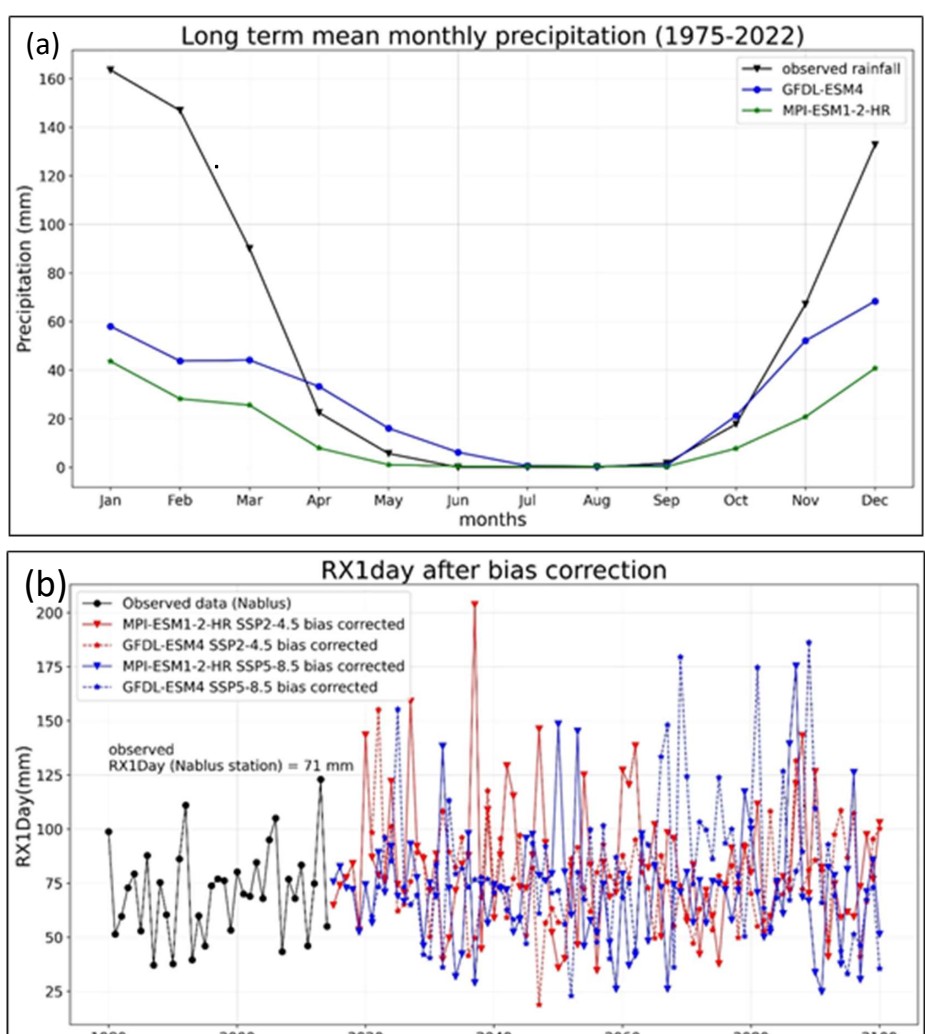

**Figure 8: Rainfall data analysis. (a) Long term mean monthly rainfall of GFDL-ESM4 and MPI-ESM1-2-HR, compared with**
**observed rainfall. (b) Historical RX1day and future RX1day (after bias correction).**

The Mann-Kendall trend test performed on bias-corrected future RX1day values indicated no significant trends, prompting the

application of stationary rainfall frequency analysis using Gumbel's method. Table 4 shows the changes in RX1day with

respect to historical data for different return periods in the mid-future period, for both GCMs and both SSPs. For both the

scenarios, GFDL-ESM4 projected a decrease in RX1day, whereas MPI-ESM1-2-HR projected an increase as high as 39%.

This difference highlights the uncertainties inherent in climate models and future climate for the region. The return period



rainfall for the near future, mid-future, and far future is shown in Figure 9. Both models showed an increase in rainfall compared to historical RX1day, except for GFDL-ESM4 in the mid-future (Figure 9b). Higher changes were observed in SSP5-8.5 for the near future, while SSP2-4.5 showed more significant changes in the far future.

**Table 4: Return period rainfall for GFDL-ESM4 and MPI-ESM1-2-HR, SSP2-4.5 and SSP5-8.5 for the mid-future.**

| Rainfall in Mid-future (2041–2060) for SSP2-4.5 | | | | | |
|---|---|---|---|---|---|
| Return Period (years) | Historical 1985–2014 (mm) | GFDL-ESM4 (mm) | % change | MPI-ESM1-2-HR (mm) | % change |
| 5 | 89 | 89 | 0.2 | 112 | 26 |
| 10 | 103 | 102 | -0.9 | 135 | 30 |
| 25 | 121 | 119 | -1.9 | 163 | 34 |
| 50 | 135 | 132 | -2.5 | 184 | 37 |
| 100 | 148 | 144 | -2.9 | 205 | 39 |
| Rainfall in Mid-Future (2041–2060) for SSP5-8.5 | | | | | |
| 5 | 89 | 86 | -2.9 | 102 | 15 |
| 10 | 103 | 100 | -2.8 | 123 | 19 |
| 25 | 121 | 118 | -2.6 | 149 | 23 |
| 50 | 135 | 131 | -2.5 | 168 | 25 |
| 100 | 148 | 145 | -2.4 | 188 | 27 |


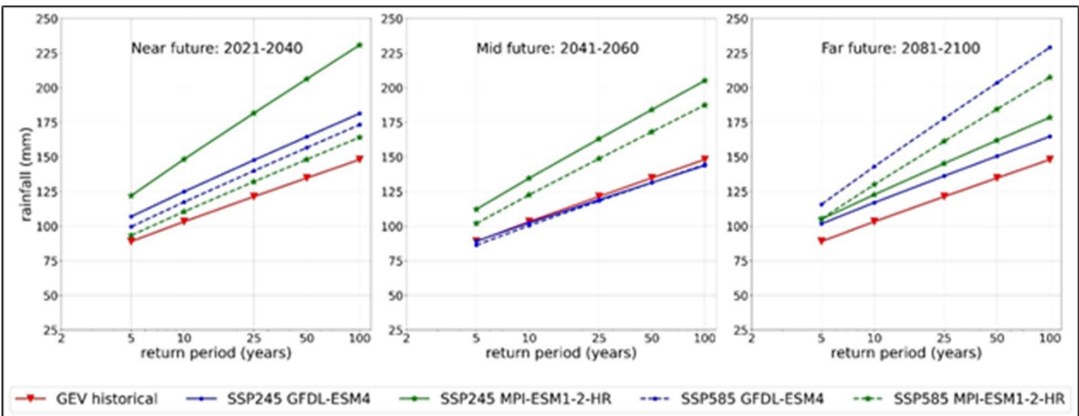

**Figure 9: Projected rainfall values for return periods and SSP2-4.5 and SSP5-8.5 for the near future, mid-future, and far future.**




### 4.2.2 Flood hazard

The rainfall recorded on the 8th of January 2013 (106.7, 89.0, and 81.5 mm of rainfall for Nablus, Tulkarm, and Jenin rain gauges respectively) corresponded to a historical return period of between 1 in 5 (89 mm) to 1 in 10 years (103 mm) (Table 4). A 1 in 100-year event based on historical data (1985–2014) would feature 148 mm of rainfall. The highest projected future

rainfall for a 1 in 100-year event (205 mm) was estimated by the mid-future (2041–2060) MPI-ESM1-2-HR model and SSP2-4.5 scenario (Table 4). Owing to the uncertainty in the future climate between the GFDL-ESM4 and MPI-ESM1-2-HR models (Table 4), we modelled both these rainfall events to quantify the impacts of higher magnitude flooding. To derive a spatially variable rainfall grid, the Nablus rainfall scenarios were scaled proportionally for Tulkarm and Jenin stations using the 2013 flood rainfall distributions (Table 5). Only FastFlood and HAIL-CAESAR were applied across the study area due to the

computational limitations of using HEC-RAS.

**Table 5: RX1day rainfall scenarios and flood hazard.**

| Event | Nablus rainfall (mm) | Tulkarm rainfall (mm) | Jenin rainfall (mm) | Inundated area (km²) |
|---|---|---|---|---|
| 2013 flood | 106.7 | 89.0 | 81.5 | 21.84 (FastFlood) 17.39 (HAIL-CAESAR) |
| Historical 1 in 100-year | 148 | 123.4* | 113.0* | 23.17 (FastFlood) 17.94 (HAIL-CAESAR) |
| 1 in 100-year mid-future (MPI-ESM1-2-HR model, SSP2-4.5 scenario) | 205 | 171.0* | 156.6* | 26.84 (FastFlood) 21.41 (HAIL-CAESAR) |

*Rainfall values were derived as a proportion of Nablus's rainfall according to the 2013 rainfall event.







**Figure 10: HAIL-CAESAR and FastFlood flood inundation for historical and future (MPI-ESM1-2-HR model, SSP2-4.5 scenario) 1 in 100-year RX1day rainfall events. Impacts are shown for the number of buildings intersecting with the inundated area, the length of transport network within the inundated area, and the area of cropland within the inundated area. Percentages are with respect to the total number, length, or area of the features for the study area. Building footprints are from Microsoft (2024), the road network is from OCHA (2021), and the cropland area is from ESA WorldCover V200 (Zanaga et al., 2021).**





FastFlood produced the largest inundated area for each 1 in 100-year rainfall scenario, with 26.84 km$^2$ inundated for the future 1 in 100-year rainfall event (Table 5). This model also inundated the greatest length of transportation network including roads and track (111.5 km or 9.2 % of the total length in the study area), and the largest area of cropland (14.5 km$^2$ or 25.3% of all cropland in the study area) (Figure 10d). However, the HAIL-CAESAR model for the same scenario inundated a larger number

of buildings (n = 3,207) (Figure 10c). Differences in the modelled inundation between FastFlood and HAIL-CAESAR were apparent in the low relief area north of Arraba town where FastFlood produced greater floodplain inundation (Figure 10).

## 5 Discussion

Strategies to reduce flood risk are becoming more urgent as the probability of extreme rainfall events increases with climate warming (Min et al., 2011; O'Gorman, 2015). In lower income countries, this is coupled with unregulated development into

hazardous areas, undeveloped infrastructure to manage flood waters, and a lack of flood hazard maps required for decision making (Hassan et al., 2010; Rentschler et al., 2022; Sampson et al., 2015; Shadeed, 2018). Reliable urban flood hazard modelling requires accurate and typically high resolution (<10 m) DEM, combined with past flood inundation data, both of which are not globally available or open access (Fewtrell et al., 2008; Hawker et al., 2018; Neal et al., 2012; Shrestha et al., 2023). Therefore, site-specific approaches are often required. In this study, we used flood extents observed in satellite

imagery to evaluate three flood models, which were underpinned by a bespoke 10 m resolution DEM and future climate projections incorporating local rain gauge observations (1985–2014).

Satellite imagery spanning the January 2013 flood event was used to derive NDVI's that captured damaged and scoured vegetation reflecting the flood extent (Figure 4). Similar approaches have been used elsewhere to observe flash flood

inundation extents (Atefi and Miura, 2022; Miles et al., 2018). However, this method does not represent a definitive flood map since not all riparian areas will experience vegetation loss or damage, and seasonal agricultural activity including crop harvest could bias the observations as was indicated in our study. Nonetheless, NDVI difference maps (Figure 4) revealed widespread flood impacts to cropland (24% of cropland in the study area) (Figure 5), which supports observations of widespread damage to crops in the northern West Bank reported by OCHA (2013). Additionally, the 2013 flooding

prompted the formation of the Palestinian Agricultural Disaster Risk Reduction and Insurance Fund (PADRRIF) to reduce agricultural damage and losses (FAO, 2017). Probabilistic flood hazard mapping is a key mechanism to enable this preparatory risk mitigation and preparedness.

In the absence of gauging station data or other flood extent observations, NDVI differencing provided the basis to evaluate

three flood models' ability to simulate the 2013 event (Table 2 and Figure 6). As expected, the physics-based flood models HEC-RAS and HAIL-CAESAR best matched the NDVI-derived flood extent, followed by FastFlood (Table 2). Similar accuracy assessment F1 scores of 0.74, 0.75, and 0.76 for FastFlood, HAIL-CAESAR, and HEC-RAS respectively showed



that all models provided a reasonable match to the NDVI-derived flood extent. Additionally, FastFlood's run time of 40 seconds for the full study area without requiring high-performance computing or complex model setup, demonstrates its

value in providing useful flood hazard information, particularly where numerical modelling resources are limited (van den Bout et al., 2023; Najafi et al., 2024; Watson et al., 2024). In this study we focused on the maximum flood extent and depths as an indicator of impact. We would expect that dynamic flood effects, including arrival time and flow velocity, would be better captured by physics-based flood models but these were not considered due to the lack of gauging station validation data.


In studying the impacts of climate change, GCM projections are a primary source of uncertainty (Teng et al., 2012), which affects the successive steps including bias correction and rainfall frequency analysis (Shrestha et al., 2023). Selecting models that accurately represent regional-scale climate is crucial for reducing uncertainty in future climate projections (Ahmadalipour et al., 2017). This study used GCMs recommended by Hamed et al., (2022) and Mesgari et al., (2022), which evaluated 11

climate models (CMIP5 and CMIP6 versions) over the MENA region and assessed the performance of 11 CMIP6 models over the MENAP region, respectively. The inter-model variation can be seen in the return period rainfall values (Table 4). GFDL-ESM4 projects mainly negative precipitation changes with respect to historical values, while MPI-ESM1-2-HR projects increases of up to 39%. Using GCMs specifically representative of Palestine's weather patterns can help reduce uncertainties. For example, the flood hazard modelling study of Shrestha et al. (2023) applied climate models representative of Nepal (warm-

dry, cold-dry, warm-wet and cold-wet conditions), determined using the envelope-based approach by Lutz et al., (2016). Similarly, Richardson, (2020) followed a process-based evaluation based on McSweeney et al. (2015), who used realistic models with maximum possible range of changes to determine the climate models generating suitable information about future changes in extreme precipitation in South Asia. Downscaling methods add to the uncertainty of future climate projections (Teng et al., 2012). Quantile mapping, which has showed better performance for bias correction of stationary data (Heo et al.,

2019) was used in this study to correct the systematic biases of the GCMs. Here, the distribution of observed data is transferred to the projected values. Therefore, the quality of observed data also influences the biases in future climate uncertainty.

The rainfall recorded on the 8th of January 2013 (106.7, 89.0, and 81.5 mm of rainfall for Nablus, Tulkarm, and Jenin rain gauges respectively) corresponded to a historical return period of between 1 in 5 (89 mm) to 1 in 10 years (103 mm), whereas

a 1 in 100-year rainfall event (RX1day: 148 mm) from historical data (1985–2014) could become 205 mm in the mid-future (2041–2060). Flood models under a mid-future precipitation scenario (MPI-ESM1-2-HR) suggested a 23% (4 km$^2$) greater inundation extent compared to the 2013 event, which could affect over 3,000 buildings and 100 km of road network (Figure 10). In comparison, OCHA (2013) reported damage to 1,570 houses during the January 213 flood event, although it is not clear if the strong winds associated with the winter storm contributed to this count. Flash flooding affects built infrastructure

and causes damage and erosion to cropland, but also presents an opportunity for water storage and groundwater recharge, which could simultaneously reduce flood hazard. Groundwater aquifers sustain populations and agricultural activity in the



West Bank but their recharge is projected to decline with climate warming (Mizyed, 2009). Issues with water quality linked to groundwater recharge are also a concern since inadequate waste water management and runoff from agricultural areas is linked to observations of increased nitrate contamination (Anayah and Almasri, 2009; Hejaz et al., 2020). Our flood hazard assessment

provides the first high-resolution mapping for the region that can support urban planning and infrastructure development to manage storm water runoff and improve water security. Whilst this analysis acts to bound a range of flood hazard scenarios under current and future climate, future climate scenarios remain uncertain in the models we evaluated.

**6 Conclusions**

In this study, we used pre- and post-flood satellite imagery from an extreme rainfall event in January 2013 to map the associated inundation extent and impacts in the northern West Bank, Palestine. These extents were used as reference data to evaluate the performance of three flood models and quantify current and future flood hazard. Climate analysis revealed that the January 2013 rainfall corresponded to a historical return period of between 1 in 5 to 1 in 10 years. The patterns of future precipitation in the region are uncertain, although more frequent precipitation extremes are likely to increase the risk of flash flooding. Our

analysis showed that a 1 in 100-year rainfall event (RX1day: 148 mm) based on historical data (1985–2014) could become 205 mm in the mid-future (2041–2060), which could cause 23% (4 km$^2$) greater inundation compared to the 2013 event. Buildings, the road network, and agricultural land are particularly susceptible to flooding and infrastructure development will be required to manage storm water runoff, particularly where channels intersect the road network. Our study demonstrates the value of high-resolution satellite observations to observe flood extents, which then supports model calibration in data scare

regions lacking other hydrological observations. Whilst the physics-based HEC-RAS flood model displayed the best performance, the FastFlood model was able to produce a similar inundation pattern and flood depths over 300 times faster using standard computing resources, which provides greater flexibility for deployment within an urban planning decision support environment.

**Data availability**

The flood model outputs of this study and supporting data will be made available at https://zenodo.org/doi/10.5281/zenodo.13583373

**Author contributions**

All authors have read and agreed to the published version of the manuscript. CSW, MC, JG, and SS, designed the concept. SS and JD provided access to datasets. DLS and RH performed the climate analysis. CSW performed the other analysis and

prepared the manuscript with input from all authors.



**Competing interests.**

The contact author has declared that neither they nor their co-authors have any competing interests.

**Acknowledgments**

We thank the Ministry of Local Government (Palestine) for providing topographic data and the Palestine Meteorological
Department for providing rainfall records. HAIL-CAESAR modelling was undertaken on ARC4, part of the High Performance
Computing facilities at the University of Leeds, UK.

**Financial support**

This research has been supported by the UK Research and Innovation (UKRI) Global Challenges Research Fund (GCRF)
Urban Disaster Risk Hub (grant no. NE/S009000/1) (Tomorrow's Cities), and the Centre for Observation and Modelling of
Earthquakes, Volcanoes and Tectonics (COMET). COMET is a partnership between UK Universities and the British
Geological Survey.

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
