# Peer review of "Earth observation informed modelling of flash floods"

_EGUsphere, 2024_

## Referee Comment (RC1)

**Review egusphere-2024-2722**

**General comments:**

In this study three hydrological models are applied to on the West-Bank of Palestine. Because of little data availability, satellite images of the inundation extent of a flash flood in 2013 are used to validate the models. Future rainfall scenarios for the West-Bank are derived from GCMs via down-scaling and then used as input for these models to investigate possible future flood scenarios.

The manuscript is well written, clearly structured and to the point. At the same time the manuscript provides enough detail to follow the methodology and deals thoroughly with the data, e.g. the use and comparison of different DEMs.

Providing solid flood risk management in regions with little data availability is an important topic, especially in regard to climate change and the expected increase of flood magnitudes. This study points out that creative solutions in these regions are required and offers the use of satellite imagery as one possible solution.

While I really like the practical and applied nature, I also have some questions and criticism to address, mainly the use of apparently uncalibrated models which are even more simplified with the use of just one (?) mannings roughness value for the whole study area.

However, after discussing following aspects, I recommend this manuscript for publication.

**Specific comments:**

- Study area: I am missing information about the land use within the study area. You are using the NDVI for the validation of the flood extent. Is the catchment mainly characterized by agricultural land use? It would be nice, if you could add a little description about the land use percentages in the study area.

- Validation via NDVI: While flood damages on crops is relevant, one main focus of flood risk management lies on the protection of urban areas and residential/industrial buildings. How does the NDVI help to identify inundated urban areas? How do you explain the NDVI change in urban areas in Figure 5? In the abstract you point out "populated urban environments" (with usually little vegetation) but your method can only detect changes in vegetation. This remains somewhat unclear throughout the manuscript and also relates to previous question. How well is the NDVI suited to detect flooding in urban environments? In my opinion you have to elaborate more on this point.

- Discussion: You could not validate the inundation depths with your approach. However, inundation depth is one main factor regarding damages. This should be discussed.

- Line 51: "NASA Global Precipitation Measurement" and antecedent soil moisture conditions via radar: I am highly skeptical about the quality of these products. Remote sensing of soil moisture is very uncertain and can just give information about the upper layers of the soil. The spatial resolution remotely sensed precipitation is too coarse to be effectively used in flash flood modelling were affected basins are usually small and spatio-temporal distribution of rainfall is decisive for the formation of flood peaks. Please discuss this, e.g. in the following sentence ("Barriers to earth-observation ....").

- Line 184: Did you really use just one mannings roughness value for the whole study area? Why? You could derive these also from land use / or satellite data. How do you justify this enormous simplification of surface runoff processes? This is one main point of my criticism.

- Which parameters are needed to run the three hydrological model? How did you derive these parameters? I understand that there is no way to calibrate the models but did you just use standard settings for the models? Please be more transparent about this and explain how you derived the parameters. While your models might be able to replicate the extent of the 2013 flood, the might fail to reproduce the correct inundation depths and flow velocities, which are both crucial parameters regarding the flood impact on buildings. This is another main point of my criticism.

- Figure 5: I find the differences between fluvial and non-fluvial intersection quite small. (Fig 5b and c)). What could be the explanation for this? I would expect, that the inundated areas at fluvial intersections are much larger.

- Table 3: It is good that you use to different climate projections as they are really uncertain. Especially the GFDL-ESM4 on the SSP5-8.5 scenario seems quite strange with a decrease in rainfall in the mid-future. Is this an artifact?

- Table 4: How did you derive the return periods? How long was the time series you based this calculation on? Please clarify.

- Line 390: What do you refer to here, if you speak about opportunities for groundwater recharge? During flash floods infiltration is negligible. Do you mean infiltration from ponded areas after the flood? Also there seems to be an issue with causality in this sentence: "Flash flooding […] could reduce flood hazard"?

**Technical comments:**

Line 28 ff.: I think there's something wrong in this sentence. How is a warming climate driven by higher magnitude flood events?

Figure 5: Why the label "Open water". You can not see it in the three figures. I wonder if a simple table would be even easier to interpret than this figure.

Figure 8 and 9: Can you increase the resolution of these plots?

Figure 10: The differences in flooding extent are very hard to see. Maybe a table with the numbers would be enough?

---

## Author Comment (AC1)

https://doi.org/10.5194/egusphere-2024-2722

egusphere-2024-2722
Title: Earth observation informed modelling of flash floods
Author(s): C. Scott Watson et al.
MS type: Research article

**Review - egusphere-2024-2722**

**We thank the reviewer for taking the time to evaluate our manuscript and highlight several important points, which we have addressed below in bold to improve our manuscript.**

Specific comments:
- Study area: I am missing information about the land use within the study area. You are using the NDVI for the validation of the flood extent. Is the catchment mainly characterized by agricultural land use? It would be nice, if you could add a little description about the land use percentages in the study area.

**The study area is primarily comprised of landcovers that would be expected to show a vegetation response to flooding. We have added a supplementary table (Table S1) showing the land covers in the study area and we now introduce this under the Study Area section with the following text.**

**‘The main landcover classes in the study area are grassland (45%), shrubland (17%), tree cover (16%), cropland (11%) and built-up (10%) (Table S1).’**

- Validation via NDVI: While flood damages on crops is relevant, one main focus of flood risk management lies on the protection of urban areas and residential/industrial buildings. How does the NDVI help to identify inundated urban areas? How do you explain the NDVI change in urban areas in Figure 5? In the abstract you point out "populated urban environments" (with usually little vegetation) but your method can only detect changes in vegetation. This remains somewhat unclear throughout the manuscript and also relates to previous question. How well is the NDVI suited to detect flooding in urban environments? In my opinion you have to elaborate more on this point.

**We agree that using a metric of vegetation change to derive flood extent in urban areas is problematic due to the built-up nature and sparser vegetation coverage, which is also why we did not create validation extents from these areas. No flood map or field evidence was available for the 2013 event. Therefore, we decided to use the NDVI-derived flood extents outside urban areas to calibrate the flood models, which were applied across the full study area. We did not intend our approach to be used to derive validation flood extents in urban areas, and validation in the urban areas is therefore lacking. We have added further discussion to clarify this:**

**Section 4.1.1**
**‘The highest stream orders displayed greater spread in the NDVI change (Figure 4d), likely due to a combination of their less ephemeral nature, greater carrying capacity, and lower detection of NDVI changes for channels in built-up environments due to sparser vegetation coverage. Therefore, using NDVI change to derive reference flood extents would not be**

*appropriate in urban areas; however, the deposition of sediment in these areas could be used instead if they had sufficient spectral contrast to the surrounding roads and buildings (Notti et al., 2018).'*

**Section 5. Discussion:**
*'We were only able to validate the models against flood extents derived outside urban areas, where the spectral change in vegetation post-flood was clear. Within built-up areas, the deposition of sediment could be used to derive information on flood extent if it had sufficient contrast with the adjacent landcovers (Notti et al., 2018). The lack of apparent deposition and supporting field observations, precluded this in our study.'*

**Roads, tracks and buildings are generally small in the study area, without large industrial complexes or concreted surfaces for example. The pixels of built-up area in the landcover map will in some cases therefore be a mix of urban structures/roads/tracks and vegetation, since the map has a 10 m pixel size. Therefore, vegetation loss or damage in these areas (see example image below) was observed and reported in the NDVI difference (Fig. 5). To better isolate specific impacts, we reported the number of buildings and length of roads and tracks within the modelled flood area (e.g. Fig. 10). We have clarified this with text under 3.2.2:**

*'The 10 m pixel size of the ESA WorldCover land cover map would in some cases incorporate mixed pixels of urban structures, roads, tracks, and vegetation for example, given that these features are generally small within the study area. Therefore, to evaluate potential inundation impacts for urban areas, we also used building footprints from the Global ML Buildings dataset (Microsoft, 2024), and the transportation network including roads and tracks from OCHA (OCHA, 2021), which were more complete than OpenStreetMap data.'*

**The example image below shows the 2013 RapidEye satellite image background, built-up area landcover (red), the areas of significant NDVI decrease (blue), and the overlap (orange). The latter (orange) would be reported as built-up area in Fig. 5b.**

https://doi.org/10.5194/egusphere-2024-2722

[Figure]

- Discussion: You could not validate the inundation depths with your approach. However, inundation depth is one main factor regarding damages. This should be discussed.

**Yes – unfortunately our project, and this paper, was working with case study locations where validation data was sparse or completely absent. Hence, a motivating factor for evaluating NDVI-derived flood extents. We recognise the importance of other factors for assessing damage to agricultural land and buildings, including flood depth and velocity, but unfortunately, without calibration data, we could not validate them here. We have modified the discussion text on this topic to read:**

*'In this study we focused on the maximum flood extent and depths as an indicator of impact. However, we recognise that we could only validate flood extent due to the absence of flood depth information and gauging station validation data for the 2013 event. We were only able to validate the models against flood extents derived outside urban areas, where the spectral change in vegetation post-flood was clear. Within built-up areas, the deposition of sediment could be used to derive information on flood extent if it had sufficient contrast with the adjacent landcovers (Notti et al., 2018). The lack of apparent deposition and supporting field observations, precluded this in our study. Additionally, dynamic flood effects, including arrival time, flow velocity, and depth, could also be better represented by physics-based flood models if gauging station validation data were available. These factors are key to more accurately assessing potential damages to agricultural land and buildings, for example (Hammond et al., 2015; Smith, 1994).'*

- Line 51: "NASA Global Precipitation Measurement" and antecedent soil moisture conditions via radar: I am highly skeptical about the quality of these products. Remote

sensing of soil moisture is very uncertain and can just give information about the upper layers of the soil. The spatial resolution remotely sensed precipitation is too coarse to be effectively used in flash flood modelling were affected basins are usually small and spatio-temporal distribution of rainfall is decisive for the formation of flood peaks. Please discuss this, e.g. in the following sentence ("Barriers to earth-observation ....").

**We agree that is problematic for smaller basins. Therefore, we showed the GPM data for our catchment but used the rainfall gauges as the primary data source (Fig. 3b). We have added the following text as suggested:**

***'Additionally, the coarse resolution global products such as GPM may not capture the spatio-temporal complexities of precipitation and therefore the formation of flood peaks in flash flooding (Sapountzis et al., 2021).'***

- Line 184: Did you really use just one mannings roughness value for the whole study area? Why? You could derive these also from land use / or satellite data. How do you justify this enormous simplification of surface runoff processes? This is one main point of my criticism.

**We understand your concern over this simplification, which is why we also tested the model sensitivity to varying manning's value chosen in Supplementary Table 1. In our catchment, the highest resolution land cover information available was the 10 m resolution ESA WorldCover 200 product and we used this data in our analysis of land cover inundation (Fig. 5). However, the channels in our catchment are less than a pixel wide, and often only active following rainfall, so are not resolved in this map (e.g. image below). Additionally, the study area is largely comprised of grassland (yellows) and shrubland (oranges), with interspersed areas mapped as trees/woodland (greens) in the landcover map (image below). In reality, these wooded areas are often olive groves, with trees regularly spaced and with mixed landcover interspersed (bare ground/grassland/shrubland). Other land covers are seasonal and we were unable to obtain observations of likely roughness. Given the uncertainty in the landcover map, and lack of appropriate information/calibration data to inform spatially variable manning's values, we instead chose to calibrate a uniform value against our reference flood extents. Using a uniform value is not uncommon in similar circumstances (Begnudelli and Sanders, 2007; Fewtrell et al., 2008; Jamali et al., 2019; Shrestha et al., 2023).**

**We have added the following text to the methods to justify the selection of a uniform manning's value and clarify that this is a simplification under section 3.4.1:**

***'Spatially variable roughness values are preferable to capture detailed inundation characteristics, for example informed by a landcover map. However, applying a uniform value is common where landcover data are not sufficiently high resolution or are uncertain, and where there is a lack of field-base information to inform and calibrate spatially variable Manning's values (Begnudelli and Sanders, 2007; Fewtrell et al., 2008; Jamali et al., 2019; Shrestha et al., 2023).'***

**Landcover map with an example of the flood extent (blue polygon) overlaid:**

https://doi.org/10.5194/egusphere-2024-2722

[Figure]

- Which parameters are needed to run the three hydrological model? How did you derive these parameters? I understand that there is no way to calibrate the models but did you just use standard settings for the models? Please be more transparent about this and explain how you derived the parameters. While your models might be able to replicate the extent of the 2013 flood, the might fail to reproduce the correct inundation depths and flow velocities, which are both crucial parameters regarding the flood impact on buildings. This is another main point of my criticism.

**Unfortunately, we were not able to validate inundation depths and we have added further discussion of this in relation to your earlier point. However, the aim of the study was to use earth observation data (NDVI change) as a means to provide at least some information (flood extent) to inform flood modelling in ungauged catchments. Whilst recognising that there are notable compromises in the approach taken due to the lack of validation data.**

**We have added text under 3.4.1 to clarify that the models use standard settings, other than the parameters we specify in Table 1.**

***'In the absence of validation data in our catchment, we used standard model parameters* and performed *sensitivity testing to Manning's roughness values, which were uniformly applied across the study area for each simulation.'***

- Figure 5: I find the differences between fluvial and non-fluvial intersection quite small. (Fig 5b and c)). What could be the explanation for this? I would expect, that the inundated areas at fluvial intersections are much larger.

**The NDVI decrease corresponding to the stream network is apparent across the study area (Fig. 4b), but so too are the large areas of NDVI decrease not directly intersecting with the stream network (Fig. 4a). This is because the precipitation from the 2013 flood event would have ponded damaged vegetation across the study site, causing NDVI changes. Some of these areas may indeed drain to lower order streams, but did not directly intersect with them in our analysis. For the higher order streams, flowing water is the most likely mechanism to remove or damage vegetation causing an NDVI response. We have clarified the discussion of this figure to read:**

***'The NDVI decrease intersecting with the stream network, which was most likely to be a direct result of the effect of moving water on the vegetation,...'.***

https://doi.org/10.5194/egusphere-2024-2722

*'Non-fluvial NDVI change represents a vegetation response to the storm precipitation or standing water, although these areas may still drain into lower order streams. Here, NDVI decrease was greatest for grassland....'*

- Table 3: It is good that you use to different climate projections as they are really uncertain. Especially the GFDL-ESM4 on the SSP5-8.5 scenario seems quite strange with a decrease in rainfall in the mid-future. Is this an artifact?

**The study area lies within a region characterized by low model agreement, particularly regarding projected changes in annual maximum daily precipitation under different Shared Socio-economic Pathways, including SSP5-8.5 scenario (Seneviratne et al., 2021). According to Ali et al. (2022), projections for south-east Mediterranean indicate a mixed direction of change and also decreases in heavy precipitation at global warming level of 1.5°C and 4°C respectively. Therefore, the decrease in RX1day projected by GFDL-ESM4 is not necessarily an artifact, but rather a plausible outcome within the range of projections assessed by the IPCC.**

Table 3: Percentage changes in future RX1day after bias correction

| Scenarios | Historical 1985–2014 | Near future 2021–2040 | % change | Mid-future 2041–2060 | % change | Far future 2081–2100 | % change |
|---|---|---|---|---|---|---|---|
| RX1day (mm) and % changes: GFDL-ESM4 | | | | | | | |
| SSP2-4.5 | 71 | 84 | 18 | 72 | 1 | 82 | 15 |
| SSP5-8.5 | | 76 | 8 | 68 | -4 | 80 | 13 |
| RX1day (mm) and % changes: MPI-ESM1-2-HR | | | | | | | |
| SSP2-4.5 | 71 | 88 | 24 | 83 | 17 | 82 | 15 |
| SSP5-8.5 | | 71 | 0 | 75 | 6 | 73 | 3 |

- Table 4: How did you derive the return periods? How long was the time series you based this calculation on? Please clarify.

**The return period values presented in Table 4 were derived through a stationary frequency analysis using Gumbel's extreme value distribution, following the steps:**

1) **30 years of annual maximum daily rainfall data for each period was used.**
2) **Historical base period from 1985-2014 and Mid-Future period from 2041-2060.**
3) **From the daily rainfall data, the annual maximum rainfall for each year was extracted — for each of 30 years, one value (the maximum daily rainfall value) was selected.**
4) **Then Gumbel's Distribution was applied, which is a commonly used method for modelling extreme events such as extreme rainfall — to fit the 30 annual maximum values.**
5) **Once the distribution was fitted, rainfall amounts corresponding to specific return periods i.e., 5-year, 10-year, 25-year, 50-year and 100-year was calculated.**

- Line 390: What do you refer to here, if you speak about opportunities for groundwater recharge? During flash floods infiltration is negligible. Do you mean infiltration from ponded areas after the flood? Also there seems to be an issue with causality in this sentence: "Flash flooding [...] could reduce flood hazard"?

**Yes –ponding was apparent following the 2013 flood and in our flood models. We have corrected this sentence to read:**

https://doi.org/10.5194/egusphere-2024-2722

*'Flash flooding affects built infrastructure and causes damage and erosion to cropland. However, ponded water, which was apparent in the 2013 flood and our simulations, presents an opportunity for groundwater recharge. Groundwater aquifers...'*

Technical comments:

Line 28 ff.: I think there's something wrong in this sentence. How is a warming climate driven by higher magnitude flood events?

**This sentence was missing the 'flood risk' context. We have split the sentence into two, and modified to read:**

*'A warming climate with more frequent extreme rainfall events (Min et al., 2011; O'Gorman, 2015) is coupled with increased exposure of populations and infrastructure to flooding (Alfieri et al., 2017; Jongman et al., 2012; Tellman et al., 2021). Flood risk is driven by factors including higher magnitude flood events (Hirabayashi et al., 2013; Slater et al., 2021a), human-modified catchment runoff characteristics (Kundzewicz et al., 2014), and encroachment into flood-prone areas (Andreadis et al., 2022; Devitt et al., 2023).'*

Figure 5: Why the label "Open water". You can not see it in the three figures. I wonder if a simple table would be even easier to interpret than this figure.

**We have removed the label 'Open water' as suggested. We appreciate your suggestion but in this case prefer the bar plot to a table in order to visualise the difference in overall NDVI increase and decrease for each plot, in addition to the contributing land covers.**

Figure 8 and 9: Can you increase the resolution of these plots?

**Yes, the resolution of the plots has now been increased.**

[Figure]

https://doi.org/10.5194/egusphere-2024-2722

[Figure]

Figure 10: The differences in flooding extent are very hard to see. Maybe a table with the numbers would be enough?

**We appreciate that the flooding extents are not easy to see across the full study area, although differences are noticeable. We wanted to include a figure that shows the modelled floods for the full study extent, along with a summary of the impacts for each scenario. Therefore, we have retained this figure. The overall inundated areas are additionally presented in Table 5. We will make the flood map Geotiff outputs available on Zenodo so they can be viewed at full resolution by readers in GIS software.**

References:

Begnudelli, L. and Sanders, B. F.: Simulation of the St. Francis Dam-Break Flood, J. Eng. Mech., 133, 1200–1212, https://doi.org/10.1061/(ASCE)0733-9399(2007)133:11(1200), 2007.

Fewtrell, T. J., Bates, P. D., Horritt, M., and Hunter, N. M.: Evaluating the effect of scale in flood inundation modelling in urban environments, Hydrol. Process., 22, 5107–5118, https://doi.org/10.1002/hyp.7148, 2008.

https://doi.org/10.5194/egusphere-2024-2722

Hammond, M. J., Chen ,A.S., Djordjević ,S., Butler ,D., and and Mark, O.: Urban flood impact assessment: A state-of-the-art review, Urban Water J., 12, 14–29, https://doi.org/10.1080/1573062X.2013.857421, 2015.

Jamali, B., Bach, P. M., Cunningham, L., and Deletic, A.: A Cellular Automata Fast Flood Evaluation (CA-ffé) Model, Water Resour. Res., 55, 4936–4953, https://doi.org/10.1029/2018WR023679, 2019.

Sapountzis, M., Kastridis, A., Kazamias, A. P., Karagiannidis, A., Nikopoulos, P., and Lagouvardos, K.: Utilization and uncertainties of satellite precipitation data in flash flood hydrological analysis in ungauged watersheds, Glob Nest J, 23, 388–399, 2021.

Shrestha, D., Basnyat, D. B., Gyawali, J., Creed, M. J., Sinclair, H. D., Golding, B., Muthusamy, M., Shrestha, S., Watson, C. S., Subedi, D. L., and Haiju, R.: Rainfall extremes under future climate change with implications for urban flood risk in Kathmandu, Nepal, Int. J. Disaster Risk Reduct., 97, 103997, https://doi.org/10.1016/j.ijdrr.2023.103997, 2023.

Smith, D. I.: Flood damage estimation - A review of urban stage-damage curves and loss functions, Water SA, 20, 231–238, https://doi.org/10.10520/AJA03784738_1124, 1994.

Ali, E., W. Cramer, J. Carnicer, E. Georgopoulou, N.J.M. Hilmi, G. Le Cozannet, and P. Lionello. 'Cross-Chapter Paper 4: Mediterranean Region. In: Climate Change 2022: Impacts, Adaptation and Vulnerability. Contribution of Working Group II to the Sixth Assessment Report of the Intergovernmental Panel on Climate Change'. Cambridge University Press, 2022. https://doi.org/10.1017/9781009325844.021.

---

## Author Comment (AC2)

https://doi.org/10.5194/egusphere-2024-2722

egusphere-2024-2722
Title: Earth observation informed modelling of flash floods
Author(s): C. Scott Watson et al.
MS type: Research article

**Review 2 - egusphere-2024-2722**

**We thank the reviewer for taking the time to evaluate our manuscript. We have addressed the points raised below in bold.**

I found the manuscript to include several interesting elements, but I struggled to grasp a clear central message. It seems to be trying to do three things at once: (1) demonstrate how NDVI-based satellite observations can be used to support flood model validation in data-scarce areas, (2) compare three flood models of different complexity in terms of their performance and efficiency, and (3) simulate future flood hazard under changing climate conditions. All of these are relevant and useful topics, but the way they are presented together in the same paper feels unfocused. It's unclear which of these is the main contribution.

The title emphasizes "Earth observation informed modelling," which suggests that the novelty lies in using NDVI (and potentially other EO data) to support flood model validation where traditional in-situ observations are unavailable. This is potentially a very valuable idea and an important contribution. However, the paper spends a lot of time comparing three models and running future scenarios, and those parts — while well executed — feel somewhat disconnected from the core innovation. If the goal is to highlight the value of NDVI as a validation tool, then that angle needs to be brought forward much more clearly, both in the framing and the discussion. If, instead, the authors are more interested in benchmarking models or demonstrating a future risk pipeline, then the paper might need a different title and narrative altogether.

As it stands, the paper tries to be a methods paper, a modelling comparison, and a climate risk study all at once — and as a result, the reader is left unsure what the takeaway is. I would encourage the authors to clarify their core message, streamline the structure around that message, and remove or reduce content that is not essential to it. A more focused version of this study could be very publishable, but in its current form I would recommend substantial revision and resubmission.

**We understand your concern that including a NDVI analysis, flood model comparison, and climate change analysis has potentially resulted in an unclear core message and takeaway from the paper. It was always our intention to include each element to demonstrate how the earth observation data (NDVI change) could support flood hazard modelling in data sparse regions, such as our chosen study site. Therefore, we used the NDVI analysis in an applied research perspective to evaluate three flood models in the absence of traditional calibration data (e.g. river gauge data). This could then inform a future flood risk assessment using a climate change analysis. We believe this integrated approach is in the scope of the journal but fully agree that we need to present a clear aim and coherent discussion throughout the paper. Focusing on one of the elements for multiple case studies, for example deriving reference flood extents using the NDVI differencing approach, would be a valuable analysis. However, in**

https://doi.org/10.5194/egusphere-2024-2722

our application in a data sparse flash flood region, we only had a single historical flood example to draw upon. Therefore, we prefer to clarify the importance of our integrated approach and remove unnecessary detail, rather than focussing the study on one element.

We propose the following changes to clarify the core message and remove non-essential content:

- We have modified our aim and objective to clarify the link between each element of the manuscript, and explicitly include mention of the NDVI differencing:
    - *'In this study, we aimed to draw on our experience in the application of the DSE in Nablus, Palestine, to evaluate how satellite data can be used to inform flood hazard modelling in data sparse flash flood regions through the development and implementation of an end-to-end methodology. Our objectives were to: (1) delineate the flood extent of a major historical flood using NDVI differencing applied to pre- and post-flood satellite imagery; (2) evaluate the performance of three flood hazard models of increasing complexity against the observed flood extent; and (3) apply the validated flood models to inform an assessment of current and future flood hazard in the region. We aimed to create and apply an end-to-end methodology to assess flood hazard.'*
- We have modified the introductory text under '3.3 Rainfall data and climate scenarios' and '3.4.1 January 2013 flooding' to clarify how these sections link to the overall analysis.
    - *'Historical and future projected rainfall data were used in a climate change analysis to demonstrate how satellite analysis of the observed historical flood event could inform a future flood hazard assessment.'*
    - *'Three models were used to simulate the January 2013 flooding and evaluate their accuracy with respect to the historical flood extent'*
- Additional text was included under '4.1.1 NDVI change' in response to Reviewer 1, which expands the discussion on the method and its limitations.
    - *'...and lower detection of NDVI changes for channels in built-up environments due to sparser vegetation coverage. Therefore, using NDVI change to derive reference flood extents would not be appropriate in urban areas; however, the deposition of sediment in these areas could be used instead if they had sufficient spectral contrast to the surrounding roads and buildings (Notti et al., 2018).'*
    - *'Non-fluvial NDVI change represents a vegetation response to the storm precipitation or standing water. These areas may still drain into lower order streams but are unlikely to be associated with main channel flooding.'*
- We have removed part of the discussion about the climate models to streamline this section:
    - **'Downscaling methods add to the uncertainty of future climate projections (Teng et al., 2012). Quantile mapping, which has showed better performance for bias correction of stationary data (Heo et al., 2019) was used in this study to correct the systematic biases of the GCMs. Here, the distribution of observed**

https://doi.org/10.5194/egusphere-2024-2722

> data is transferred to the projected values. Therefore, the quality of observed data also influences the biases in future climate uncertainty.'

- We have moved Figure 8, the rainfall data and bias correction, to the supplement, since this supports the climate analysis but the detail is not required in the main text.
- We have modified text in the conclusion to mention the NDVI differencing approach:
  - *'In this study, we used pre- and post-flood satellite imagery from an extreme rainfall event in January 2013 to map the associated inundation extent and impacts in the northern West Bank, Palestine using an NDVI differencing approach'.*
  - *We have reordered and modified the text to finish the conclusion with:* 'Our study demonstrates the value of high-resolution multi-spectral satellite observations to derive flood extents through NDVI differencing following a flash flood, which then supports model calibration in data scare regions lacking other hydrological observations such as gauging stations, or where post-event mapping of flood characteristics is not available.'